# Structural basis of nucleic-acid recognition and double-strand unwinding by the essential neuronal protein Pur-alpha

Janine Weber[1], Han Bao[2], Christoph Hartlmüller[3], Zhiqin Wang[2], Almut Windhager[1], Robert Janowski[1], Tobias Madl[1,3,4,5], Peng Jin[2], Dierk Niessing[1,6]*

[1]Institute of Structural Biology, Helmholtz Zentrum München - German Research Center for Environmental Health, Neuherberg, Germany; [2]Department of Human Genetics, Emory University, Atlanta, United States; [3]Center for Integrated Protein Science Munich, Department of Chemistry, Technische Universität München, Munich, Germany; [4]Institute of Molecular Biology & Biochemistry, Center of Molecular Medicine, Medical University of Graz, Graz, Austria; [5]Omics Center Graz, BioTechMed Graz, Graz, Austria; [6]Department Cell Biology, Biomedical Center of the Ludwig-Maximilians-University München, Planegg-Martinsried, Germany

**Abstract** The neuronal DNA-/RNA-binding protein Pur-alpha is a transcription regulator and core factor for mRNA localization. Pur-alpha-deficient mice die after birth with pleiotropic neuronal defects. Here, we report the crystal structure of the DNA-/RNA-binding domain of Pur-alpha in complex with ssDNA. It reveals base-specific recognition and offers a molecular explanation for the effect of point mutations in the 5q31.3 microdeletion syndrome. Consistent with the crystal structure, biochemical and NMR data indicate that Pur-alpha binds DNA and RNA in the same way, suggesting binding modes for tri- and hexanucleotide-repeat RNAs in two neurodegenerative RNAopathies. Additionally, structure-based in vitro experiments resolved the molecular mechanism of Pur-alpha's unwindase activity. Complementing in vivo analyses in *Drosophila* demonstrated the importance of a highly conserved phenylalanine for Pur-alpha's unwinding and neuroprotective function. By uncovering the molecular mechanisms of nucleic-acid binding, this study contributes to understanding the cellular role of Pur-alpha and its implications in neurodegenerative diseases.

*For correspondence: dierk.
niessing@med.uni-muenchen.de

**Competing interests:** The authors declare that no competing interests exist.

## Introduction

Purine-rich element-binding protein A (Pur-alpha) plays a crucial role in postnatal brain development. Pur-alpha-deficient mice appear normal at birth but develop severe neurological abnormalities after 2 weeks and die shortly after birth (*Hokkanen et al., 2012*; *Khalili et al., 2003*). These mice show fewer cells in the brain cortex, hippocampus, and cerebellum as a consequence of decreased proliferation of the precursor cells. Further studies indicated that Pur-alpha co-localizes with Staufen and FMRP and that Pur-alpha [(-/-)] mice display dendritic mislocalization of both proteins (*Johnson et al., 2006*). In support of its important neuronal function, point mutations in the human Pur-alpha gene have been found to cause the so-called 5q31.3 microdeletion syndrome, which is characterized by neonatal hypotonia, encephalopathy, and severe developmental delay (*Lalani et al., 2014*; *Hunt et al., 2014*; *Tanaka et al., 2015*).

Pur-alpha is an ubiquitously expressed, multifunctional protein that binds to both DNA and RNA and is known to regulate replication, transcription, and translation (*Johnson et al., 2013*). It has been shown that Pur-alpha binds to single- and double-stranded nucleic acids that contain GGN

**eLife digest** Some proteins perform several different tasks inside cells. This is the case for a protein called Pur-alpha, which is essential for neurons to work correctly. For example, Pur-alpha can bind to DNA to regulate gene activity. It also binds to RNA molecules, which are copies of a gene, and helps to distribute them within the neuron. In humans, there are several neurodegenerative diseases in which Pur-alpha is involved. One example is the Fragile X-associated Tremor/Ataxia Syndrome (FXTAS), which causes memory and movement problems.

Experiments with isolated proteins and double-stranded DNA show that Pur-alpha is able to separate the two DNA strands. But it was not clear how this DNA unwinding occurs, and the biological significance of this activity was unknown. Other questions also remained unanswered: how does Pur-alpha recognize DNA and RNA? Does the loss of Pur-alpha's binding to DNA and RNA contribute to neurodegenerative diseases?

To address these questions, Weber et al. obtained Pur-alpha from the fruit fly and crystallized the protein bound to DNA. A technique called X-ray crystallography was then used to determine the three-dimensional structure of the Pur-alpha/DNA complex in fine enough detail to work out the position of individual atoms.

Based on this structure, Weber et al. could introduce mutations that alter the DNA- and RNA-binding region of the protein to investigate the binding mechanism. The crystal structure and experiments with normal and mutant Pur-alpha protein revealed how it unwinds double-stranded DNA: binding of Pur-alpha to DNA causes a strong twist of the DNA molecule, which contributes to separating the strands. Further experiments in fruit flies revealed that both the DNA-unwinding activity and the ability of Pur-alpha to bind DNA/RNA are needed for the protein to work correctly in neurons.

Because Pur-alpha is involved in a range of different processes inside cells, a future goal is to identify the DNA and RNA sequences it specifically binds to. This information, together with the insights gained from Weber et al.'s study, should advance our understanding of why Pur-alpha is essential for maintaining neurons.

motifs. Such regions are found at origins of DNA replication and enhancers of TATA-box lacking genes, such as *c-myc* or the myelin-basic protein, which Pur-alpha regulates. Pur-alpha has also been routinely purified from cytoplasmic kinesin-containing ribonucleoprotein particles (RNPs) (*Kanai et al., 2004*; *Ohashi et al., 2000*), further supporting its role in mRNA localization and showing that Pur-alpha is a core factor in localizing mRNPs.

Besides its ability to bind RNA and DNA, Pur-alpha possesses dsDNA-destabilizing activity in an ATP-independent fashion (*Darbinian et al., 2001*). This function has been suggested as important for DNA replication and transcription regulation. It was postulated that Pur-alpha, being a transcription activator, contacts the purine-rich strand of promoter regions and displaces the pyrimidine-rich strand, which would allow the binding of other proteins and activation of transcription (*Darbinian et al., 2001*; *Wortman et al., 2005*). The role of Pur-alpha-dependent unwinding activity in RNA localization and in RNA-based neuropathological disorders is currently unknown.

One particularly interesting interaction partner of Pur-alpha is the RNA helicase Rm62, the *Drosophila* ortholog of p68. It is implicated in transcriptional regulation, pre-mRNA splicing, RNA interference, and nucleo-cytoplasmic shuttling (*Qurashi et al., 2011*). Thus, their joint function could be the initial unwinding of short dsRNA regions by Pur-alpha followed by helicase-dependent melting of larger regions for the regulation of RNA processing, translational control, and transport.

Nucleic acid-binding of Pur-alpha is mediated by three central PUR repeats (*Graebsch et al., 2010*; *Graebsch et al., 2009*), which are N-terminally flanked by unstructured, glycine-rich sequences and C-terminally by glutamine- and glutamate-rich regions (*Figure 1A*; *Johnson et al., 2013*). In the recently published crystal structure of Pur-alpha each of both PUR repeats I and II consist of a four-stranded antiparallel beta-sheet, followed by a single alpha-helix (*Graebsch et al., 2009*). Repeat I and II fold into an intramolecular dimer that serves as a DNA-/RNA-binding domain. The third repeat leads to intermolecular dimerization (*Figure 1A*; *Graebsch et al., 2009*). Despite these

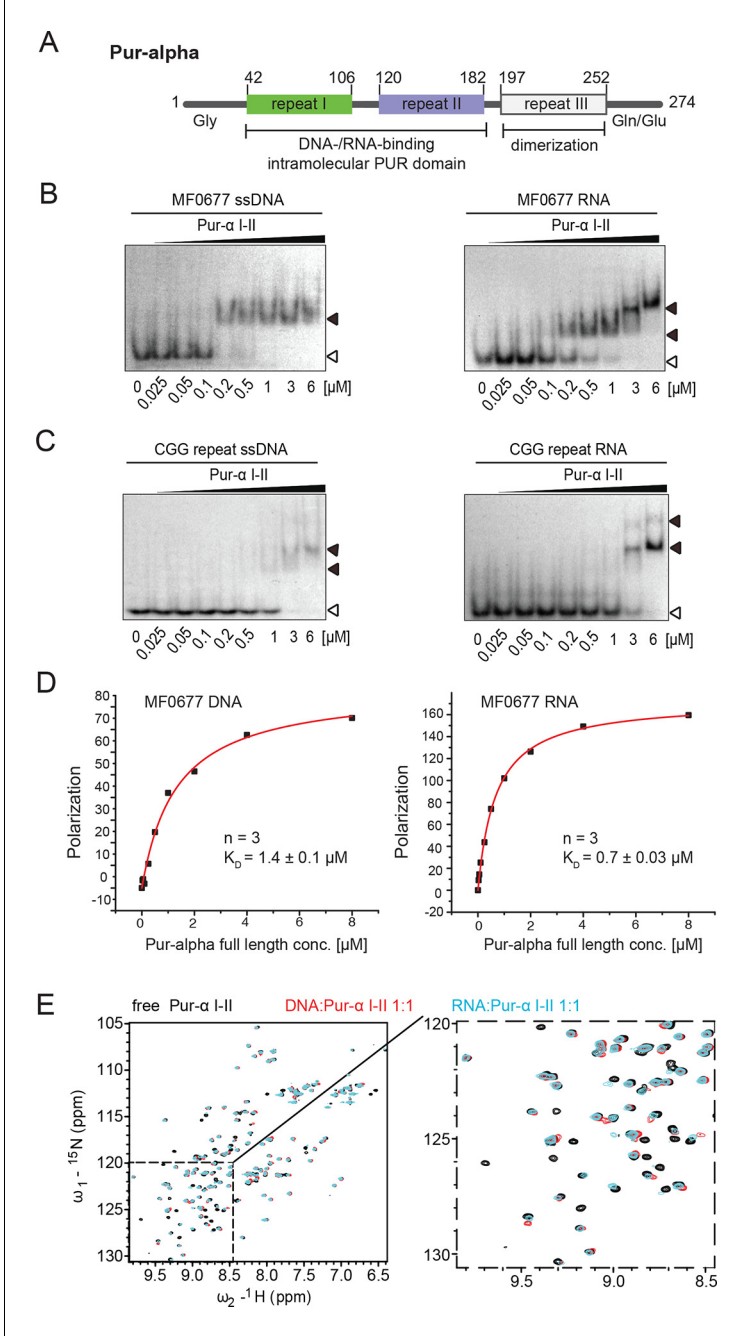

**Figure 1.** Pur-alpha uses similar binding modes for DNA and RNA. (**A**) Schematic representation of the *Drosophila* Pur-alpha protein, comprising 274 amino acids. Cartoon shows PUR repeat I (green) and II (blue), forming the intramolecular DNA-/RNA-binding PUR domain, and PUR repeat III (grey) mediating dimerization. The N-terminal unstructured, glycine-rich region and the C-terminal glutamine-/glutamate-rich region are indicated by Gly and Gln/Glu, respectively. Numbers indicate amino-acid positions of domain boundaries. (**B, C**) Radioactive EMSA with *Drosophila* Pur-alpha repeat I-II. (**B**) Pur-alpha repeat I-II binds to MF0677 ssDNA (left) and ssRNA (right) with similar affinities. (**C**) Pur-alpha repeat I-II binds to CGG-repeat ssDNA (left) and RNA (right) also with similar affinity, but less strong than to the MF0677 sequence. Open arrowheads indicate free and filled arrowheads indicate protein-bound DNA/RNA oligonucleotides. (**D**) Fluorescence-polarization experiments with full-length Pur-alpha and nucleic acids. The full-length protein shows a twofold stronger binding to MF0677 ssRNA when compared to MF0677 ssDNA. (**E**) Binding of unlabeled GCGGA ssDNA and ssRNA to [15]N-labeled Pur-alpha repeat I-II (50 µM) followed by NMR spectroscopy. (Left) Overlay of [1]H,[15]N HSQC NMR spectra of free (black), DNA-bound (red, 1:1 ratio) and RNA-bound (cyan, 1:1 ratio) protein, respectively. (Right) Close-up on the dashed area with the same color code.

The following figure supplements are available for figure 1:

**Figure supplement 1.** Purification and quality control of *Drosophila* Pur-alpha protein derivatives used in this study.

*Figure 1 continued on next page*

*Figure 1 continued*

**Figure supplement 2.** $^{1}$H,$^{15}$N HSQC NMR spectra showing NMR titrations of $^{15}$N-labeled Pur-alpha repeat I-II (50 μM) with increasing amounts of unlabeled GCGGA ssDNA and RNA, respectively.

insights, it remains unclear how Pur-alpha interacts with its nucleic-acid targets to mediate its cellular functions. Furthermore, the mechanistic basis and physiological importance of its unwinding activity remains unresolved.

Pur-alpha has been implicated in two so-called RNA repeat-expansion diseases, which have been the focus of a number of recent studies. The first one contains expansions in the well-studied *fmr1* gene. Individuals with 55 to 200 CGG repeats, termed pre-mutation, develop the neurodegenerative Fragile X-associated Tremor/Ataxia Syndrome (FXTAS) (*Hagerman et al., 2001*), whereas healthy individuals have less than 54 trinucleotide CGG repeats in their 5'-UTR region (*Oostra and Willemsen, 2009*). It is generally accepted that expression of *FMR1* mRNA with abnormal trinucleotide-repeat expansions are the main cause of FXTAS. The second Pur-alpha related disease is caused by repeat expansions of $G_4C_2$-hexanucleotides in the first intron of the *c9orf72* transcript. These repeat expansions are considered as the most common genetic abnormality in amyotrophic lateral sclerosis (ALS) and familial frontotemporal lobal degeneration (FTLD) (*Stepto et al., 2014*). The diseases associated with both types of repeat expansions are accompanied by the formation of repeat RNA-containing protein inclusions (*Sareen et al., 2013*; *Stepto et al., 2014*,; *Xu et al., 2013*), suggesting sequestration of proteins as potential mechanism of pathology. Pur-alpha is incorporated into the inclusions of both types of disease and associates directly with the repeat RNAs (*Jin et al., 2007*,; *Xu et al., 2013*; *Rossi et al., 2015*). In fly and mouse models, the overexpression of Pur-alpha can overcome repeat-dependent neurodegeneration of both diseases (*Jin et al., 2007*,; *Xu et al., 2013*), suggesting a direct contribution of Pur-alpha to neuropathology.

Expression of 95 CGG repeats in human neuroblastoma-derived SK-N-MC cells not only induced the formation of nuclear inclusions but also impairs the architecture of the nuclear laminar and activates DNA repair-associated histone variants (*Hoem et al., 2011*). The expression of $G_4C_2$-repeat expansions cause nuclear trafficking defects, which contribute to neurotoxicity in ALS/FTLD (*Freibaum et al., 2015*; *Jovicic et al., 2015*; *Zhang et al., 2015*). Recent studies also showed that repeat-associated non-AUG (RAN) translation occurs from CGG- as well as from $G_4C_2$-repeat RNAs and that the resulting proteins can form cytoplasmic aggregates, potentially contributing to pathology (*Mori et al., 2013*; *Todd et al., 2013*). It is likely that a combination of RNA toxicity and RAN-derived protein aggregates contribute to the full manifestation of FXTAS.

Here, we used NMR chemical shift titrations together with in vitro-binding assays to demonstrate that the nucleic acid-binding domain of Pur-alpha binds RNA and DNA in the same manner. We present the co-crystal structure of Pur-alpha with a CGG trinucleotide-repeat DNA, providing a detailed structural explanation for nucleotide recognition. Pur-alpha interacts with this single-stranded DNA fragment in a sequence-specific manner with guanines and additional contacts to the phosphordiester backbone. The observed binding mode of Pur-alpha also explains its interaction with $G_4C_2$-hexanucleotide repeats. Mutational analyses as well as determination of the complex stoichiometry confirm that the DNA-/RNA-binding domain of Pur-alpha has two nucleic acid-binding sites. The structure also revealed that a highly conserved phenylalanine causes disruption of the normal base stacking and leads to a strong torsion of the DNA strand, which plays a central role in Pur-alpha's dsDNA-unwinding activity. In vivo analyses of mutant proteins reveal that nucleic-acid binding and unwinding studied in vitro are both essential for Pur-alpha's function in vivo. This information together with the crystal structure of its C-terminal dimerization domain allows us to propose a mechanism of how full-length Pur-alpha binds and unwinds dsDNA regions.

## Results

### Pur-alpha binds RNA and DNA with similar affinities

In order to assess if Pur-alpha has a binding preference for ssDNA or ssRNA, we performed electrophoretic mobility shift assays (EMSA) with the nucleic acid-binding domain of *Drosophila* Pur-alpha, consisting of repeats I-II (PUR repeat I-II; *Figure 1A*; *Figure 1—figure supplement 1A, B*) and radio-labeled DNA or RNA oligonucleotides (24 nt) of identical sequence. The MF0677 sequence was chosen as a physiological Pur-alpha target found upstream of the human *c-myc* gene (*Haas et al., 1993*; *Haas et al., 1995*). In addition, we used a CGG-repeat sequence because Pur-alpha binds to these repeats in the 5′UTR of the *FMR1* mRNA upon incorporation into FXTAS inclusions (*Jin et al., 2007*; *Sofola et al., 2007*). In these EMSA, the affinity for the physiological Pur-alpha target MF0677 is much higher ($K_D$ ~200 nM) than for the disease-related CGG-repeat sequence ($K_D$ ~2 μM) (*Figure 1B,C*; $K_D$ estimated from EMSA). However, the binding affinities for ssDNA and ssRNA of the same sequence showed no major differences.

Since full-length Pur-alpha contains a third PUR repeat, which mediates its dimerization, and additional N- and C-terminal sequences (*Figure 1A*), we also compared DNA and RNA binding of full-length Pur-alpha (*Figure 1—figure supplement 1E*). For quantification of the nucleic acid-binding affinity, we performed fluorescence-polarization experiments. Full-length Pur-alpha showed a two-fold preference in binding to MF0677 ssRNA ($K_D$ = 0.7 μM) over MF0677 ssDNA ($K_D$ = 1.4 μM; *Figure 1D*). Thus, sequences outside PUR repeats I-II seem to moderately affect nucleic-acid binding.

### NMR titration experiments reveal indistinguishable modes of RNA and DNA binding

For a more comprehensive, residue-resolved comparison of ssDNA and ssRNA binding, we performed NMR chemical shift titration experiments with $^{15}$N-labeled *Drosophila* Pur-alpha repeat I-II (*Figure 1—figure supplement 1C*) and short unlabeled GCGGA (5 nt) DNA and RNA fragments. The $^{1}$H,$^{15}$N HSQC NMR spectrum of Pur-alpha alone shows well separated cross peaks (*Figure 1E*; *Figure 1—figure supplement 2A, B*), indicating that the protein is correctly folded. Addition of either ssDNA or ssRNA resulted in almost identical, well-localized chemical shift perturbations of backbone and sidechain amide protons (*Figure 1E*; *Figure 1—figure supplement 2A, B*). Most NMR signals of residues involved in binding disappeared upon addition of DNA/RNA, thus pointing toward an intermediate exchange regime, which is characteristic for binding affinities in the high nanomolar to micromolar range. In summary, the NMR titration experiments indicate identical binding modes of PUR repeat I-II for ssDNA and for ssRNA involving the same residues in both cases.

### Crystal structure of Pur-alpha repeat I-II in complex with CGG-repeat DNA

In order to obtain high-resolution structural information of Pur-alpha binding to nucleic acids, we performed co-crystallization experiments of Pur-alpha repeat I-II with either CGG-repeat DNA or RNA. Crystals of Pur-alpha repeat I-II with a GCGGCGG trinucleotide-repeat ssDNA diffracted to a resolution of 2.0 Å. The structure was solved by molecular replacement and refined to $R_{work}$ and $R_{free}$ of 16.3% and 21.5%, respectively (*Table 1*).

The DNA-bound protein shows the typical intramolecular dimer with two PUR repeats tightly intertwined with each other, forming a globular PUR domain (*Figure 2A*; *Video 1*; *Figure 2—figure supplement 1A*; *Graebsch et al., 2009*). Each PUR repeat consists of a N-terminal four-stranded antiparallel beta sheet followed by an alpha helix. A superposition of the previously published Pur-alpha repeat I-II apo-structure (PDB ID 3K44) (*Graebsch et al., 2009*) with the structure of the protein-DNA co-complex showed only a root-mean-square deviation (RMSD) of atomic positions of 1.14 Å (*Figure 2—figure supplement 1B*). When a flexible loop region from residues L107 to K120 was excluded, the RMSD improved to 0.83 Å. Thus, no major conformational changes occur in the PUR domain upon nucleic-acid binding, which is consistent with the results obtained from NMR chemical shift titrations.

**Table 1.** Data collection/processing and refinement statistics (molecular replacement) for the two crystal structures of *Drosophila* Pur-alpha repeat I-II/DNA co-complex and Pur-alpha repeat III alone.

| | Pur-alpha repeat I-II/DNA | Pur-alpha repeat III |
|---|---|---|
| **Data collection/processing** | | |
| PDB ID | 5FGP | 5FGO |
| Beamline | ESRF ID23-2 | ESRF ID14-1 |
| Wavelength (Å) | 0.8726 | 0.9334 |
| Detector Distance (mm) | 265.433 | 261.345 |
| Number of images | 144 | 180 |
| Oscillation range (°) | 2.5 | 1.0 |
| Space group | $P2_12_12$ | $P\,1\,2_1\,1$ |
| **Cell dimensions** | | |
| *a, b, c* (Å) | 81.9, 40.2, 48.8 | 61.5, 55.5, 67.8 |
| α, β, γ (°) | 90.0, 90.0, 90.0 | 90.0, 95.7, 90.0 |
| Resolution (Å) | 50.0-2.0 (2.05-2.0) | 50-2.6 (2.67-2.6) |
| $R_{sym}$ or $R_{merge}$ | 12.5 (79.3) | 11.1 (68.0) |
| $I\,/\,\sigma I$ | 18.85 (2.61) | 10.4 (1.97) |
| Completeness (%) | 99.4 (94.3) | 96.5 (98.4) |
| Redundancy | 13.1 (7.6) | 1.9 (1.9) |
| **Refinement** | | |
| Resolution (Å) | 41.9-2.0 | 47.7-2.6 |
| No. reflections | 11,349 | 14,001 |
| $R_{work}\,/\,R_{free}$ | 16.3 / 21.5 | 20.6 / 28.9 |
| **No. atoms** | | |
| Protein | 1207 | 3144 |
| Ligand/ion | 145 | - |
| Water | 126 | 112 |
| *B*-factors | | |
| Protein | 24.8 | 12.5 |
| Ligand | 30.4 | - |
| Water | 35.2 | 9.55 |
| **R.m.s. deviations** | | |
| Bond lengths (Å) | 0.01 | 0.01 |
| Bond angles (°) | 1.25 | 1.36 |
| **Ramanchandran plot** | | |
| Allowed (%) | 96.0 | 93.0 |
| Additionally allowed (%) | 3.3 | 6.5 |
| Disallowed (%) | 0.7 | 0.5 |

Values in parentheses are for highest-resolution shell.

## Pur-alpha repeat I-II has two binding sites for DNA

In the crystal structure, the DNA molecule 1 (DNA 1) is clamped between residues of PUR repeat I and II (*Figure 2A, B*; *Video 1*). PUR repeat II binds DNA 1 with the residue K138 of its β-sheet, and residues N140 and R142 of the short linker (*Figure 2B, C*), whereas PUR repeat I contacts the DNA 1 via residues Q52, S53, and K54 in its short linker (*Figure 2B, D*). Pur-alpha mainly binds to stacking guanine bases, but also to one of the cytosines (C5) and to the sugar phosphate backbone (*Figure 2B*).

Within the crystal lattice the first two bases (G1 and C2) of the 5'-end of DNA 1 are base pairing with the 5'-end of the symmetry related DNA molecule (DNA 1'; *Figure 2—figure supplement 2*). The cytosine C5 in the middle of the DNA 1 strand is twisted and does not stack with the

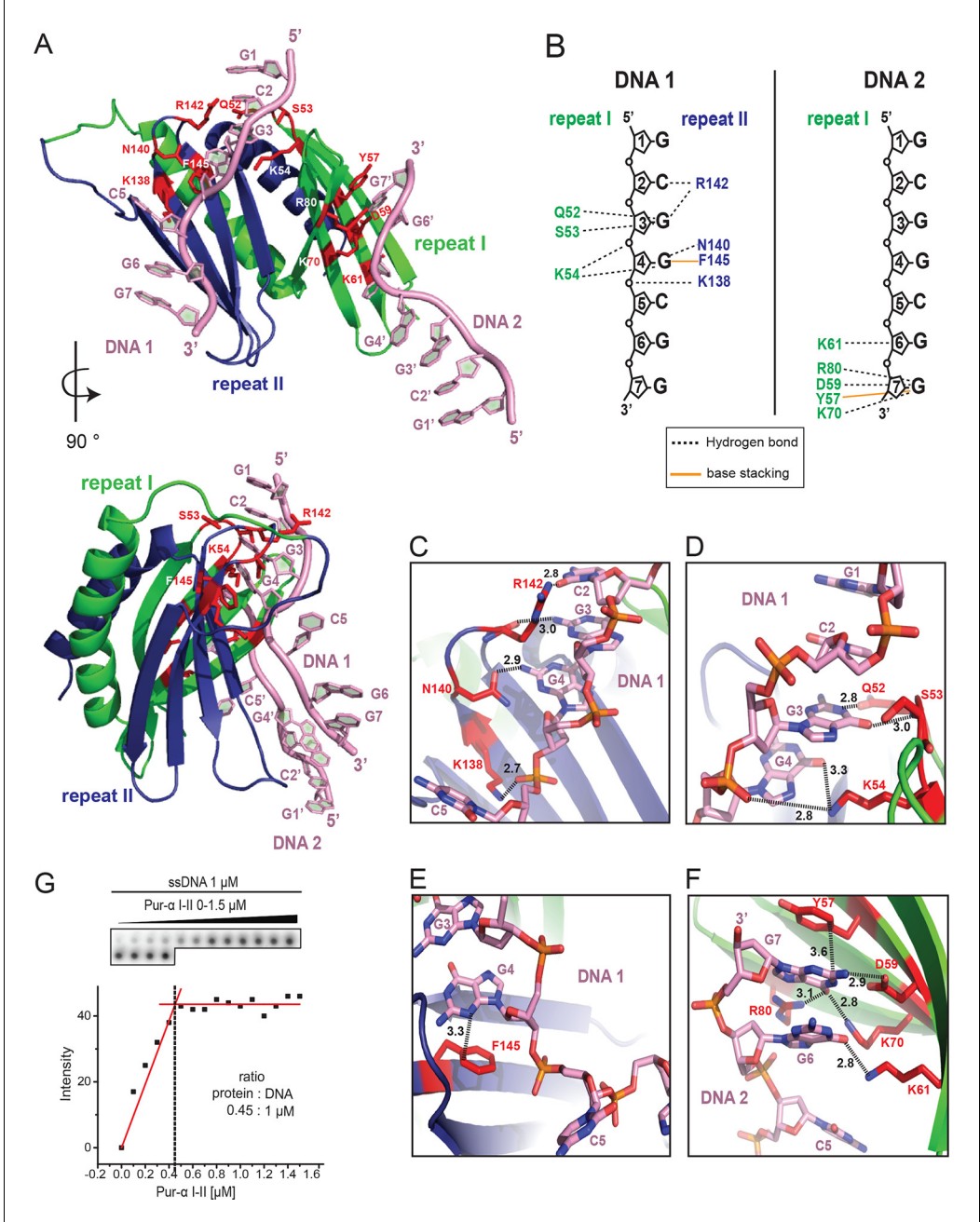

**Figure 2.** Crystal structure of *Drosophila* Pur-alpha repeat I-II in complex with the GCGGCGG ssDNA reveals that one molecule of Pur-alpha repeat I-II can bind two molecules of ssDNA. (**A**) Cartoon representation of the backbone model of the DNA-/RNA-binding domain formed by PUR repeat I (green) and II (blue) in complex with two DNA molecules (pink). Important protein residues involved in DNA interactions are depicted in red with side chains. (**B**) Schematic representation of Pur-alpha interaction with DNA molecules 1 and 2. Both PUR repeats are involved in DNA binding. Pur-alpha mainly binds to guanine bases, but also to one cytosine and the sugar phosphate backbone. (**C–F**) Detail of the protein-DNA interaction sites. (**G**) Nitrocellulose filter (top) from binding assays showing the titration of Pur-alpha repeat I-II to a constant amount of MF0677 ssDNA. The measured intensities from the filters were quantified. The graph (bottom) shows intensities from one representative binding assay, with the concentration of saturation marked with a dashed line. Three independent filter-binding assays yielded a mean saturation of 1 : 0.58 ± 0.1 µM (ssDNA : Pur-alpha repeat I-II).

The following figure supplements are available for figure 2:

**Figure supplement 1.** Analysis of the structural model of *Drosophila* Pur-alpha repeat I-II in complex with DNA.

*Figure 2 continued on next page*

*Figure 2 continued*

**Figure supplement 2.** Within the crystal structure the protein-bound DNA anneals with another symmetry-related DNA molecule.

**Figure supplement 3.** The DNA-binding site consisting of K138, N140, and R142 (KNR II) on PUR repeat II has its equivalent at the positions K61, N63, and R65 on PUR repeat I (KNR I).

**Figure supplement 4.** Amino acid sequence alignment of Pur-alpha.

neighboring guanines (*Figure 2E*). Instead, F145 from the β-sheet of PUR repeat II stacks with the neighboring guanine G4 and thereby blocks the space for the cytosine C5 (*Figure 2E*, *Video 1*).

In the crystal structure, an additional DNA-binding event was observed for PUR repeat I. The residues Y57, D59, K61, K70, and R80 of the β-sheet interact with the 3'-end of the second DNA molecule (DNA 2) (*Figure 2B, F*; *Video 1*). This interface is similar but not identical to the DNA 1-binding site on PUR repeat II. The three DNA-contacting amino acids K138, N140, and R142 of PUR repeat II are also found in corresponding positions of PUR repeat I (*Figure 2—figure supplement 3*). However, in PUR repeat I only K61 but not N63 or R65 contact the DNA 2 molecule. Thus, although there is a conservation of DNA-contacting residues on both PUR repeats, in the crystal structure their modes of binding are not identical. This observation hints toward a potentially asymmetric binding of nucleic acids on both protein surfaces of Pur-alpha I-II.

To test if Pur-alpha also interacts with two DNA oligonucleotides in solution, we performed filter-binding assays with Pur-alpha repeat I-II and MF0677 ssDNA (24 nt). Pur-alpha repeat I-II was titrated at near-stoichiometric concentrations to a constant amount (1 μM) of radiolabeled DNA and blotted onto a nitrocellulose membrane (*Figure 2G*). Plots of the signal intensities against the protein concentrations yielded a mean saturation at 0.58 ± 0.1 μM (n=3) of Pur-alpha (*Figure 2G*). This indicates a stoichiometric ratio of 1:2 (protein:DNA) and confirms that like in the crystal structure (*Figure 2A*) Pur-alpha repeat I-II binds two molecules of ssDNA in solution.

## Conserved surface patches in Pur-alpha repeat I-II contribute to DNA and RNA binding

All amino acids involved in DNA binding within the crystal structure (*Figure 2B*) are conserved (*Figure 2—figure supplement 4A*). To assess the importance of these contacts in solution, we generated structure-guided mutations and tested their effect on DNA/RNA binding. The binding motif consisting of K138, N140, R142, and F145 on PUR repeat II (KNR II and F II, respectively) is also found on PUR repeat I (K61, N63, R65, and F68; KNR I and F I, respectively). Hence, these residues were replaced by alanines and tested for nucleic acid-binding in vitro. For the QSK I – KNR II mutant the residues Q52, S53, K54, were replaced by glycine and the residues K138, N140, R142 by alanines, since a pure alanine mutant tended to aggregate. Correct folding of all generated Pur-alpha mutants was verified by circular dichroism (CD) spectroscopy (*Figure 1—figure supplement 1B*).

First, radioactive EMSA were performed with CGG-repeat and MF0677 DNA/RNA oligomers (24 nt). Except for Pur-alpha mutant F I, all other mutants showed decreased binding to DNA and RNA oligonucleotides with both motifs (*Figure 3A–E, G*; *Figure 3—figure supplement 1A–E*). In order to quantify these interactions, we performed fluorescence-polarization experiments with fluorescein-labeled MF0677 DNA and different variants of Pur-alpha. The effects observed in EMSA of mutations in Pur-alpha I-II were confirmed by these experiments (*Figure 3H*; *Figure 3—figure supplement 2*).

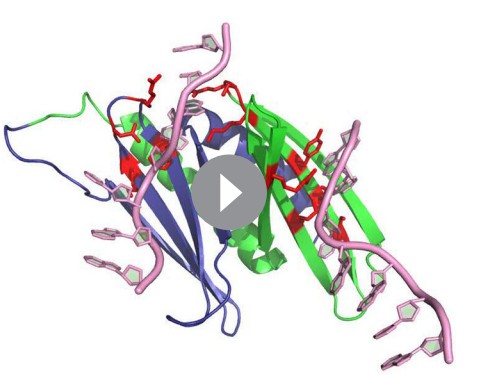

**Video 1.** Movie of the crystal structure of *Drosophila* Pur-alpha repeat I-II in complex with the GCGGCGG ssDNA. Color-coding as in *Figure 2A*. Movie relates to *Figure 2A*.

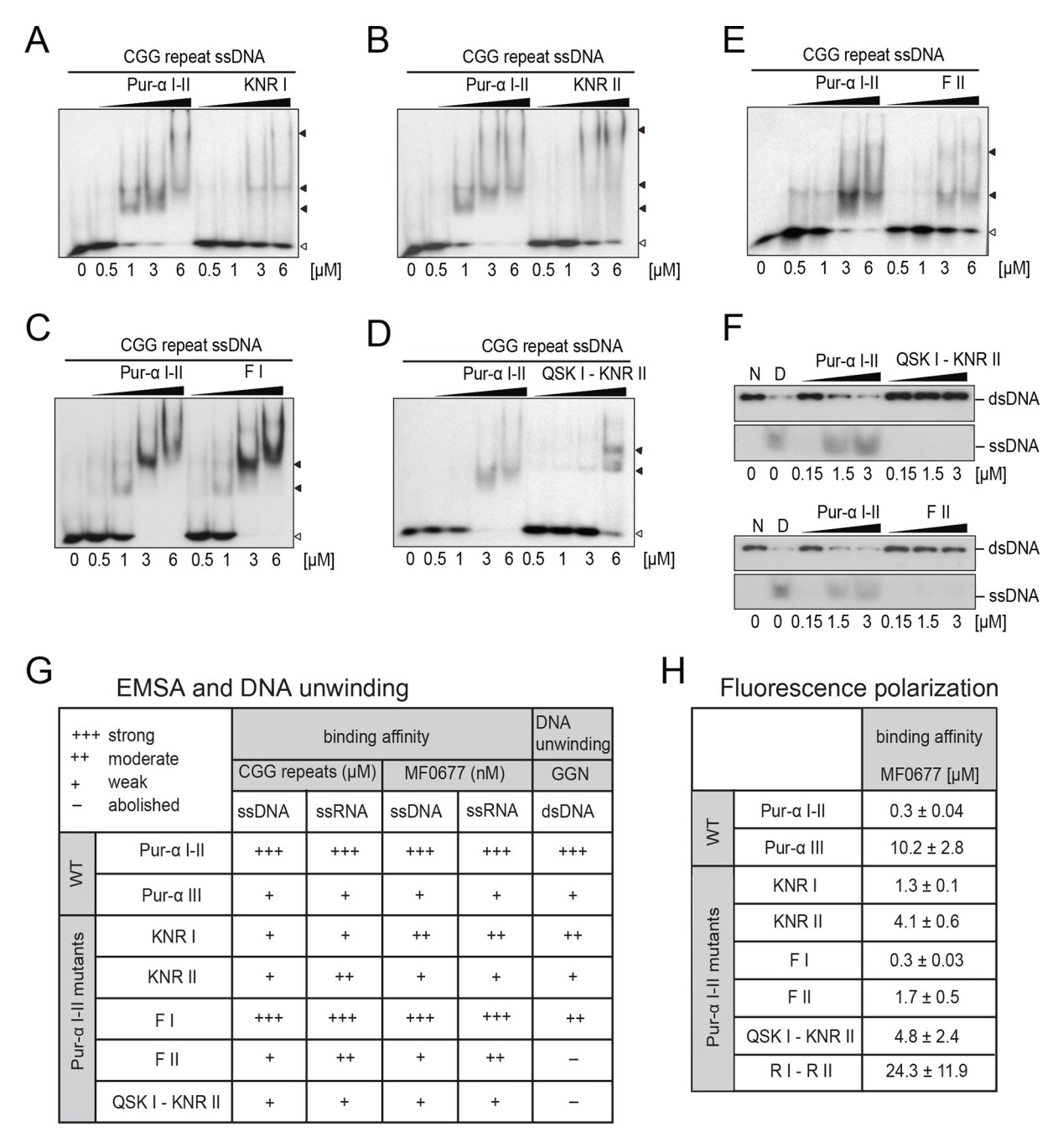

**Figure 3.** Mutations in Pur-alpha repeat I-II decrease nucleic-acid binding and dsDNA unwinding. (A–E) Radioactive EMSA with wild-type and mutant Pur-alpha repeat I-II. All mutants show a decrease in binding affinity, except for the F I mutant in C. Open arrowheads indicate free and filled arrowheads indicate protein-bound DNA/RNA oligonucleotides. (F) Unwinding assays with wild-type and mutant Pur-alpha repeat I-II. Protein was titrated to a dsDNA substrate containing a GGN motif. Pur-alpha repeat I-II is able to separate the DNA strands, whereas mutations in both repeats (QSK I – KNR II) (top) and the F II mutation (bottom) abolish the unwinding activity. (G) Summary of the results of all EMSA and unwinding experiments of Pur-alpha derivatives and mutants. Original data are shown in *Figure 3A–F* and *Figure 3—figure supplement 1*. (H) Summary of the results of fluorescence-polarization experiments of Pur-alpha derivatives and mutants with MF0677 ssDNA. Original data are shown in *Figure 3—figure supplement 2*.

The following figure supplements are available for figure 3:

**Figure supplement 1.** *Drosophila* Pur-alpha repeat I-II mutants show decreased binding affinity to DNA and RNA and decreased dsDNA-unwinding activity.

*Figure 3 continued on next page*

*Figure 3 continued*

**Figure supplement 2.** Fluorescence-polarization measurements with wild type or various mutants of Pur-alpha I-II and MF0677 ssDNA.

Of note, mutations in PUR repeat I (KNR I, F I) had less severe effects on DNA binding than mutations in repeat II (KNR II, F II).

## Phenylalanine F145 is required for unwinding activity

A large portion of the ssDNA 1 strand in the co-complex is stabilized in its conformation by aromatic stacking of G1, C2, G3, G4 and G6, G7 (*Figure 2C–F*). F145 of Pur-alpha shows particularly unusual characteristics by undergoing aromatic stacking with G4 (*Figure 2E*). This protein-DNA interaction blocks additional DNA-base stacking events and forces the DNA to flip out its cytosine base (C5), leading to a strong twist of the DNA 1 strand.

It was previously reported that Pur-alpha unwinds short stretches of dsDNA in an ATP-independent manner (*Darbinian et al., 2001*). However, the molecular basis of this function has not been understood to date. Since the sequence-specific interactions of Pur-alpha with DNA and the aromatic stacking of DNA with F145 seem incompatible with binding to dsDNA, we wondered which interactions are of foremost importance for the unwinding of dsDNA. Using a previously described unwinding assay (*Darbinian et al., 2001*), we compared ATP-independent unwinding activity of wild-type and mutant Pur-alpha repeat I-II on a dsDNA substrate.

When the main binding sites on PUR repeat I and II were mutated (QSK I – KNR II) unwinding was abolished (*Figure 3F, G*), most likely due to impaired DNA binding (*Figure 3D, G*; *Figure 3—figure supplement 1A*). In contrast, mutation of F145 (F II) abolished the unwinding activity without a complete loss of DNA binding (*Figure 3E–G*; *Figure 3—figure supplement 1E*). All other mutations showed reduced DNA binding (*Figure 3A–C, G*; *Figure 3—figure supplement 1B–D*) and only decreased unwinding (*Figure 3—figure supplement 1F*). Together these observations suggest that the heterotypic stacking of DNA-bases with F145 in PUR repeat II stabilizes the single-stranded conformation of DNA and enforces a twist of the bases that is important for its unwinding activity.

## Structure of PUR repeat III reveals a distinct function

To understand the role of the third repeat of Pur-alpha (*Figure 1A*; *Figure 1—figure supplement 1D*) for DNA/RNA binding, we determined its crystal structure. Initial datasets were obtained from native crystals at 2.7 Å resolution, from which electron-density maps were calculated by molecular replacement with the apo-structures of Pur-alpha from *Borrelia* and *Drosophila* as search templates (PDB-IDs: 3NM7 and 3K44, respectively). The final structure model was obtained in the same way from selenomethionine-derivatized crystals at 2.6 Å resolution (*Table 1*; *Figure 4A*; *Figure 4—figure supplement 1*). The structure consisting of two repeat III molecules shows the same overall fold as repeat I-II with an RMSD of 1.5 Å, and only few differences in the amino acid composition of its putative nucleic-acid-binding surface (*Figure 4A*; *Figure 2—figure supplement 4B*).

PUR repeat III was previously suggested to mainly mediate dimerization of Pur-alpha (*Graebsch et al., 2009*). However to date, no binding of PUR repeat III to nucleic acids has been measured. We therefore performed EMSA and observed that Pur-alpha repeat III bound with weaker affinities to CGG repeats and to MF0677 than Pur-alpha repeat I-II (*Figures 3G* and *4B*). Also in fluorescence-polarization experiments, PUR repeat III bound MF0677 ssDNA over 30-times weaker than PUR repeat I-II (*Figure 3H*; *Figure 3—figure supplement 2*). The main DNA/RNA interactions of full-length Pur-alpha might therefore occur via the first two PUR repeats.

Although Pur-alpha repeat III does not have a phenylalanine in the corresponding position of F145 of PUR repeat II, it also contains a conserved aromatic residue (Y219), which could potentially undergo stacking with DNA bases and support dsDNA unwinding (*Figure 2—figure supplement 4B*). However, in unwinding assays almost no activity was observed for PUR repeat III (*Figures 3G* and *4C*). These observations confirm that PUR repeats I-II mediate the main nucleic-acid-binding and unwinding activities and suggest that repeat III might predominantly mediate dimerization.

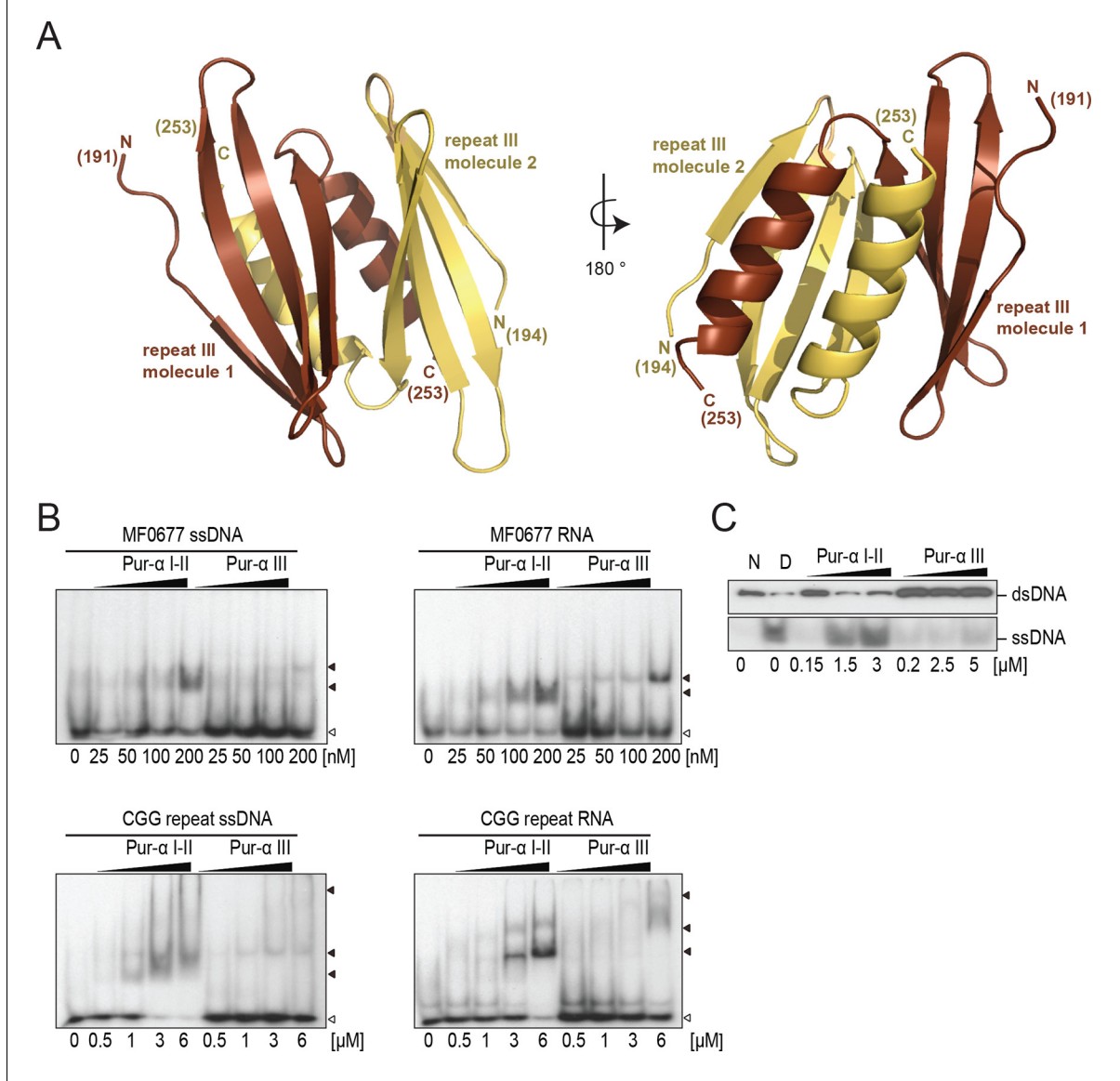

**Figure 4.** Crystal structure of PUR repeat III and assessment of its weak nucleic-acid-binding and unwinding activity. (**A**) Crystal structure of *Drosophila* Pur-alpha repeat III. Two molecules (one depicted in brown, the other in yellow) of repeat III form a dimer with intertwined α-helices, very similar to Pur-alpha repeat I-II. (**B**) Radioactive EMSA with PUR repeat III and the MF0677 DNA/RNA (top) and the CGG DNA/RNA oligonucleotides (bottom). Repeat III shows only weak binding affinity to each of both sequences, regardless of whether they consist of DNA or RNA. Open arrowheads indicate free and filled arrowheads indicate protein-bound DNA/RNA oligonucleotides. (**C**) Pur-alpha repeat III shows only weak dsDNA-unwinding activity compared to PUR repeat I-II.

The following figure supplement is available for figure 4:

**Figure supplement 1.** Analysis of the structural model of *Drosophila* Pur-alpha repeat III.

## Neuroprotection by Pur-alpha in FXTAS model requires its nucleic-acid-binding and unwinding activities

To assess the physiologic relevance of our in vitro findings, we relied on a previously reported *Drosophila* model. Overexpression of CGG-repeat RNA in the *Drosophila* eye induces neuronal degeneration and as a consequence the rough eye phenotype (compare *Figure 5A* with *5B*; *Jin et al., 2003*). Overexpression of Pur-alpha can rescue the eye phenotype in a dose-dependent manner, suggesting that this protein is sequestered into the inclusions (*Jin et al., 2007*). We compared the

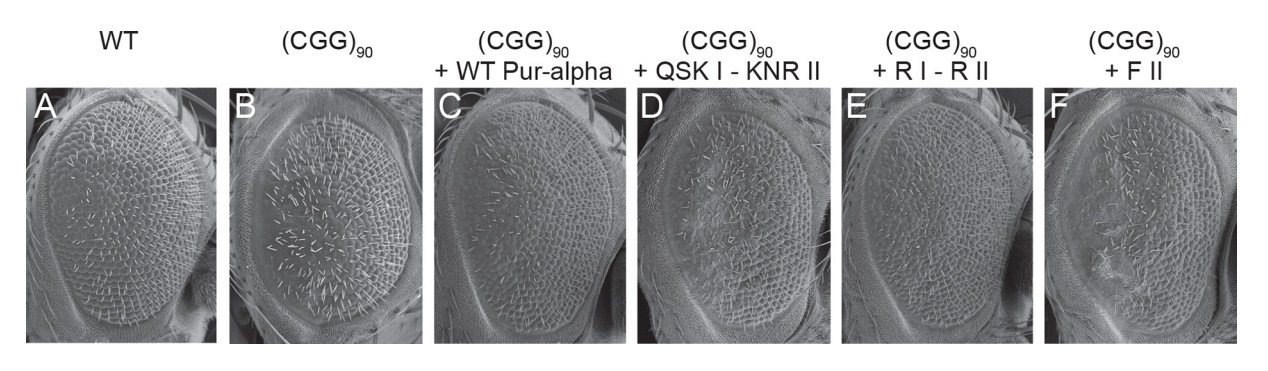

**Figure 5.** Mutations in Pur-alpha's nucleic-acid-binding domain abolish rescue of CGG RNA-mediated neurodegeneration. (A–F) Scanning electron microscope pictures of the eyes of adult flies. (A) Wild-type fly, (B) flies expressing $(CGG)_{90}$-EGFP/+ alone, together with wild-type Pur-alpha (C), with the QSK I – KNR II mutant (D), with the previously published R I – R II mutant (E) (*Graebsch et al., 2009*) and with the F II mutant (F). Only wild-type Pur-alpha and the R I – R II mutant can rescue the neurodegeneration induced by rCGG repeats.

rescue by wild-type Pur-alpha with DNA-/RNA-binding and unwinding mutants. Whereas the wild-type protein achieved a full rescue (*Figure 5C*), expression of the QSK I – KNR II mutant failed to ameliorate the rCGG repeat-induced neuronal toxicity (*Figure 5D*). On the other hand, a previously reported double mutant R80A/R158A (R I – R II) that impairs nucleic-acid binding (*Figure 3H*; *Figure 3—figure supplement 2*; *Graebsch et al., 2009*) was still able to suppress the rCGG repeat-mediated toxicity (*Figure 5E*). Thus, there might be differences in the binding for ssCGG repeats and the requirements for neuronal rescue. Most interestingly, however, is the observation that also the mutant F II, which still binds DNA/RNA but fails to unwind dsDNA, is unable to rescue neurodegeneration (*Figure 5F*). Together these observations confirm the physiologic importance of the nucleic-acid-protein contacts observed in the crystal structure. In addition, these findings formally establish that the binding and unwinding of nucleic acids is required to modulate toxicity caused by pathogenic CGG RNA.

## Discussion

Pur-alpha repeat I-II shows strong and specific binding to its physiological target MF0677 DNA located upstream of the *c-myc* gene (*Bergemann et al., 1992*), but much weaker binding to CGG-repeat RNA (*Graebsch et al., 2009*). For this reason, it has been suggested that the binding of Pur-alpha to DNA is stronger than to RNA and, as a consequence, that there might be differences in the binding modes to both nucleic-acid targets. In this study, we directly compared Pur-alpha binding to RNA and DNA oligonucleotides of the same sequence and found no major differences (*Figure 1B–D*). This suggests that the higher affinity for MF0677 ($K_D$ ~200 nM; *Figure 1B*) over CGG repeats ($K_D$ ~2 μM; *Figure 1C*) is due to differences in sequence and not the absence of the 2' OH group in the DNA. This interpretation found further support from NMR titrations with $^{15}$N-labeled Pur-alpha repeat I-II and oligonucleotides. The spectra showed similar chemical shift perturbations, regardless of whether it was DNA or RNA, indicating that both nucleic acids are bound in the same way (*Figure 1E*). Finally, the crystal structure of the Pur-alpha/DNA co-complex showed that a hydroxyl-group on the 2' position of the pentose ring of the RNA sugar backbone would not cause steric clashes (*Figure 2A,C-F*). Together, our biochemical, NMR, and X-ray crystallographic insights indicate that Pur-alpha binds DNA and RNA in the same way and thus will interact equally with both types of nucleic acids in the cell. It is also consistent with the previously suggested Pur-alpha-dependent gene regulation by competitive RNA binding (*Tretiakova et al., 1998*).

Previous findings implied that the positively charged β-sheets mediate DNA/RNAbinding, whereas the amphipathic helices might contribute to protein-protein interactions (*Graebsch et al., 2009*). The crystal structure of the protein-DNA co-complex confirms that the β-sheets, together with their short linkers, are involved in DNA binding, in contrast to the α-helices that show no interaction (*Figure 2A*). A comparison of the Pur-alpha repeat I-II apo-structure (PDB ID 3K44) with the

co-structure presented here revealed no significant conformational changes (*Figure 2—figure supplement 1B*).

In the crystal structure, Pur-alpha interacts with nucleic acids by clamping them between its two repeats, mostly by interacting with the guanine bases (*Figure 2A, B*). Only R142 interacts with the cytosine base C2. K54 and K138 additionally stabilize the DNA binding by interacting with the sugar phosphate backbone of guanine G4 and cytosine C5, respectively (*Figure 2B–D*). Binding therefore occurs sequence specifically and confirms the GGN-binding motif postulated before (*Bergemann and Johnson, 1992*).

Mutation of the interacting residues resulted in a decreased binding affinity (*Figure 3G, H*) and therefore confirmed the interaction sites seen in the crystal structure. Also mutation of the corresponding KNR motif on PUR repeat I (KNR I) caused a decrease in affinity (*Figure 3G, H*). However, in fluorescence-polarization experiments, the mutation of KNR I had a less severe effect on DNA binding ($K_D$ = 1.3 μM) than mutation of KNR II ($K_D$ = 4.1 μM; *Figure 3H*). This is consistent with the observation that in the crystal structure all three residues of KNR II make contacts with DNA 1 (*Figure 2B, C*), whereas in KNR I only a single amino acid binds to DNA 2 (*Figure 2—figure supplement 3*). Also, the F I mutation in repeat I had a less severe effect on MF0677 ssDNA binding than the F II mutation (*Figure 3G, H*). In summary, these observations suggest that the MF0677 ssDNA is bound asymmetrically by PUR repeats I-II.

In FXTAS patients, Pur-alpha binds to CGG-repeat expansions that cause the formation of nuclear inclusions and neurodegeneration (*Oostra and Willemsen, 2003*). Pur-alpha is also incorporated into inclusion triggered by $G_4C_2$-repeat RNA of patients with ALS and FTLD. The nucleic-acid binding of Pur-alpha observed in the crystal structure can explain both binding events, as it makes sequence-specific interactions with a GGC motif found in both repeat RNAs.

The structural model of Pur-alpha repeat I-II forming a PUR domain has two nucleic-acid-binding surfaces. PUR repeats I and II share the identical binding motif (KNR), and adopt the same fold, despite moderate sequence identity of about ~30% (*Figure 2—figure supplement 4*; *Graebsch et al., 2010*; *Graebsch et al., 2009*). Consistent with this finding, we observed a stoichiometric ratio of 1:2 for the PUR domain with ssDNA in filter-binding assays (*Figure 2G*). Both binding events appear at overlapping but non-identical surface regions (*Figure 2A, B*), which might prefer different GGN-motifs (GGA, GGG, GGC, GGT) as has been previously suggested (*Aumiller et al., 2012*). This might also explain why CGG repeats bind less strongly to Pur-alpha than the MF0677 sequence, which mostly consists of GGA and GGT motifs.

Pur-alpha has been previously reported to unwind dsDNA in an ATP-independent manner (*Darbinian et al., 2001*; *Wortman et al., 2005*). However, so far, it has not been shown how unwinding is achieved on a molecular level and that this function is physiologically relevant. The crystal structure of our Pur-alpha/DNA co-complex offers a mechanistic explanation: phenylalanine in position 145 of PUR repeat II undertakes base stacking with the guanine G4 and thereby blocks the space for the neighboring cytosine C5 (*Figure 2E*). Thereupon, the cytosine flips out and the 3'-end of the DNA strand becomes distorted. The interaction of K54 and K138 with the phosphate backbone upstream of the cytosine C5 enforces this strong turn (*Figure 2B–D*). F145 is highly conserved throughout different species (*Figure 2—figure supplement 4A*) and its mutation (F II) abolishes unwinding of dsDNA (*Figure 3F, G*).

Phenylalanine 145 has its structural counterpart in PUR repeat I in position F68. Although F68 is also highly conserved, in the crystal structure the guanine base stacking is not mediated by this residue. Instead, the conserved Y57 in repeat I stacks with G7 (*Figure 2B, F*). As mentioned before, the two binding sites of Pur-alpha seen in the crystal structure are asymmetric and might account for sequence-specific binding to nucleic acids with different GGN motifs.

To assess the physiological importance of the interactions observed in the crystal structure and validated in vitro, we used the previously reported FXTAS fly model (*Jin et al., 2007*; *Jin et al., 2003*). Expression of pre-mutation CGG-repeat RNA in *Drosophila* induces neurodegeneration, which is easily detectable in abnormalities in the facet eye (compare *Figure 5A* with *5B*). While we observed that overexpression of wild-type Pur-alpha rescues the eye phenotype (*Figure 5C*), the RNA-binding mutant QSK I - KNR II failed to do so (*Figure 5D*). Surprisingly, a second, previously published RNA-binding mutant (Pur-alpha R I – R II), which showed strongly reduced MF0677 ssDNA binding (*Figure 3H*), was able to fully rescue the eye phenotype (*Figure 5E*). This observation indicates that arginine 80 and 158 are not required for the binding to nucleic acids important for

neuroprotection. While the neuroprotection by the R I – R II mutant indicates flexibility in nucleic-acid recognition, the loss of rescue by the QSK I - KNR II mutant formally establishes the requirement of nucleic-acid binding for Pur-alpha-dependent neuroprotection. Additionally, the F II mutation of Pur-alpha, which abolishes its dsDNA-unwinding activity, also impairs the neuroprotective function in the fly model (*Figure 5F*). These findings indicate that unwinding is important for neuroprotection by Pur-alpha.

Recently, de novo mutations in Pur-alpha have been found to cause the so-called 5q31.3 microdeletion syndrome. This disease is characterized by neonatal hypotonia, encephalopathy, and severe developmental delay (*Lalani et al., 2014*; *Hunt et al., 2014*; *Tanaka et al., 2015*). Of the reported mutations (*Figure 6—source data 1*), two missense mutations (A89P, K97E) are of particular interest from a structure-to-function point of view (*Lalani et al., 2014*). Sequence alignment of Pur-alpha from different species shows that the residues A89 and K97 of the human Pur-alpha protein correspond to the residues A72 and R80 of the *Drosophila* protein, respectively. These residues are highly conserved (*Figure 2—figure supplement 4A*). In the crystal structure of the protein/DNA co-complex, A72 does not directly interact with the DNA molecule. Instead it forms backbone hydrogen bonds between the β-strands of PUR repeat I to stabilize the nucleic-acid binding β-sheet (*Figure 6A*, top) (this study and *Graebsch et al., 2009*). When A72 and its disease-causing counterpart A98 in the human protein (*Figure 6A*, middle) are substituted by a proline, the backbone interactions that stabilize the β-sheet very likely become disrupted (*Figure 6A*, bottom) and thus the protein misfolds.

The *Drosophila* equivalent R80 of the disease-associated human K97 directly binds to the guanine base G7 (*Figure 2B, F* and *6B*, top) and its mutation results in reduced nucleic-acid binding (*Graebsch et al., 2009*). It is therefore conceivable that a mutation of K97 to glutamate impairs nucleic-acid interaction because of repulsive forces and causes dysfunction of Pur-alpha (*Figure 6B*, middle, bottom). Although in our fly model the double mutant R80A/R158A (R I – R II) was still able to rescue neurodegeneration (*Figure 5E*), the reported effect of the K97E mutation in the microdeletion syndrome indicates that nucleic-acid binding by this residue is important at least in humans. Additional interesting disease-causing point mutations in human Pur-alpha are I188T and I206F (*Figure 6—source data 1*), which likely impair the intramolecular dimerization of PUR repeats I and II (*Hunt et al., 2014;*, *Tanaka et al., 2015*). Taken together, the crystal structure of the Pur-alpha/DNA co-complex presented in this study provides a molecular explanation for the effects of missense mutations in the 5q31.3 microdeletion syndrome.

Wild-type Pur-alpha binds to origins of replication and promoter regions (*Bergemann and Johnson, 1992,*; *Bergemann et al., 1992*) and regulates the transcription of more than 20 genes (*White et al., 2009*). Pur-alpha's ability to unwind dsDNA might therefore play an important role in the initiation of replication and transcription. One recently reported interaction partner of Pur-alpha that might play a role in this context is the RNA helicase Rm62 (*Qurashi et al., 2011*). In the light of the dsDNA-unwinding activity an intriguing speculation is that Pur-alpha also unwinds dsCGG-repeat RNA. This initial unwinding by Pur-alpha could allow interacting helicases to subsequently regulate RNA processing, transport, and translation. Therefore, it will be important to assess Pur-alpha's role in unwinding of dsRNA and its interaction with Rm62.

Considering that Pur-alpha repeat I-II has two nucleic-acid-binding sites, it is conceivable that each PUR repeat binds to one of the strands of a duplex DNA molecule thereby unwinding short stretches of dsDNA (*Figure 7A–C*, top). The insertion of Pur-alpha between both DNA strands might be achieved through spontaneous breathing of the dsDNA helix (*Peyrard et al., 2009,*; *Jose et al., 2012*). Intercalating residues (phenylalanine, tyrosine) might cause further separation of the two DNA strands via base stacking with the guanines and thereby causing the strong twist of the DNA strands. The partly melted duplex DNA could then be further unwound by DNA helicases, which are required for initiation of transcription and replication. In the crystal structure, base pairing is observed between the 5'-G1-C2 bases of two symmetry-related DNA molecules (*Figure 2—figure supplement 2*), indicating, that a PUR domain would unwind a short stretch of approximately four to six bases.

We also solved the crystal structure of PUR repeat III (*Figure 4A*; *Figure 4—figure supplement 1*) and found that it binds only weakly to DNA/RNA (*Figure 3G, H*) and unwinds dsDNA only slightly (*Figure 4C*). Since in the crystal structure the C-terminal end of PUR repeat I-II is located on the opposite side of its nucleic-acid-binding surface (*Figure 7A, B*, bottom), it is unlikely that PUR repeat

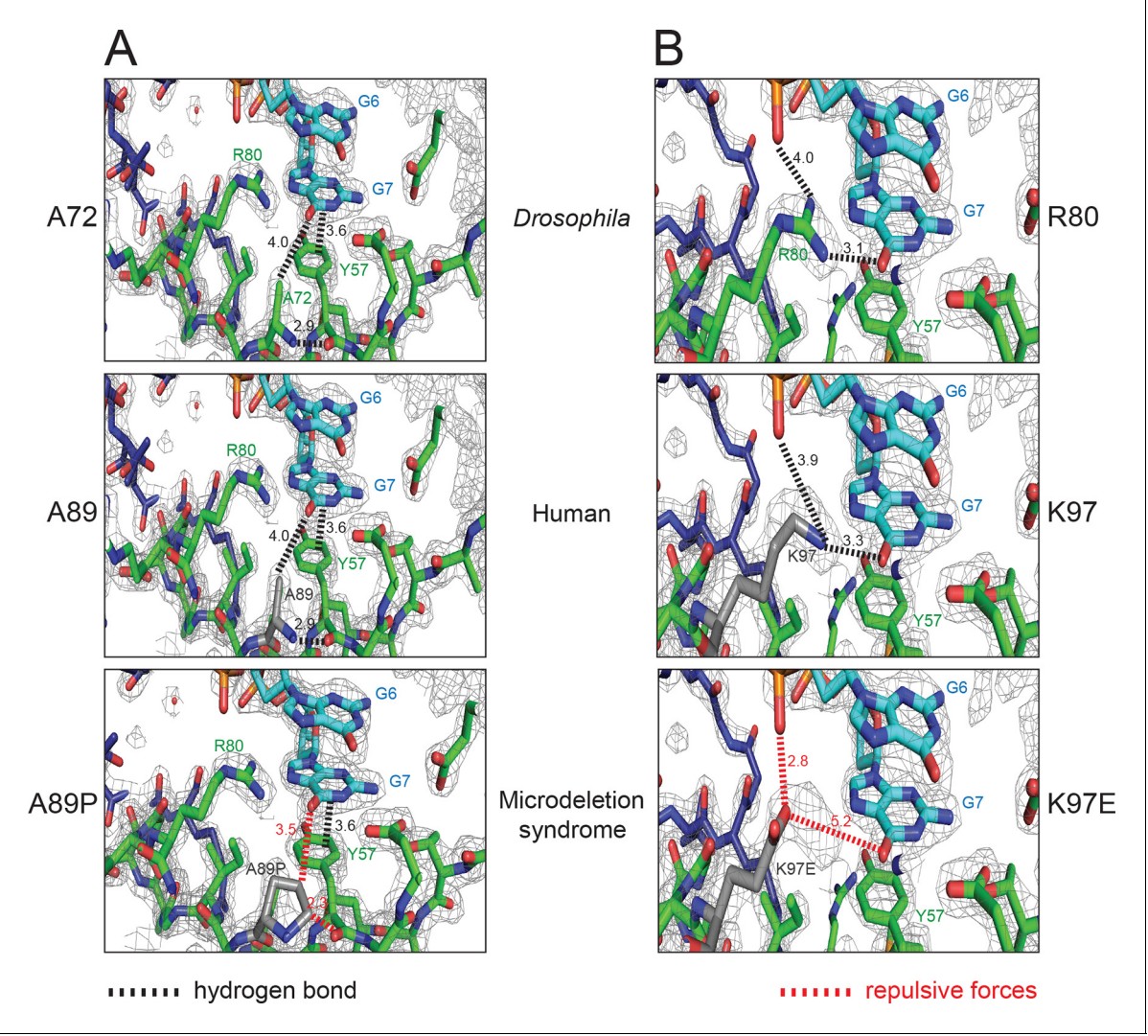

**Figure 6.** Pur-alpha mutations found in the 5q31.3 microdeletion syndrome can be modeled into the crystal structure of *Drosophila* Pur-alpha repeat I-II (green) in complex with DNA (cyan). (**A**) Residue A72 of the *Drosophila* protein (top) corresponds to the residue A89 (grey) of the human protein modeled into the *Drosophila* crystal structure (middle). In both species, the alanines form backbone hydrogen bonds. In the microdeletion syndrome A89 is mutated to proline, which disrupts backbone interactions (bottom). (**B**) Residue R80 of the *Drosophila* protein (top) corresponds to the residue K97 (grey) of the human protein, which was modeled into the crystal structure (middle). Both R80 and K97 are positively charged residues. In *Drosophila* R80 interacts with the guanine G7. The same interaction is likely to be mediated by K97. In the microdeletion syndrome, K97 is mutated to a glutamate, which probably impairs nucleic-acid binding due to its negative charges (bottom). A list of all published mutations in human Pur-alpha leading to the 5q31.3 microdeletion syndrome is shown in *Figure 6—source data 1*. In this table, their predicted effects on the structure and function of Pur-alpha are also indicated.

The following source data is available for figure 6:

**Source data 1.** Mutations in the gene encoding for human Pur-alpha that result in the 5q31.3 microdeletion syndrome.

III causes steric clashes interfering with the nucleic-acid binding by PUR repeat I-II. Hence, PUR repeat III might only facilitate dimerization, thereby guiding a second DNA-/RNA-binding domain (PUR repeat I-II) to another GGN motif further upstream or downstream on the dsDNA, where additional DNA-unwinding events could take place (*Figure 7C*, bottom). How this effect of dimeric Pur-alpha is achieved on a molecular level and if unwinding of longer dsDNA fragments requires its joint action with helicases are main questions to be addressed in future.

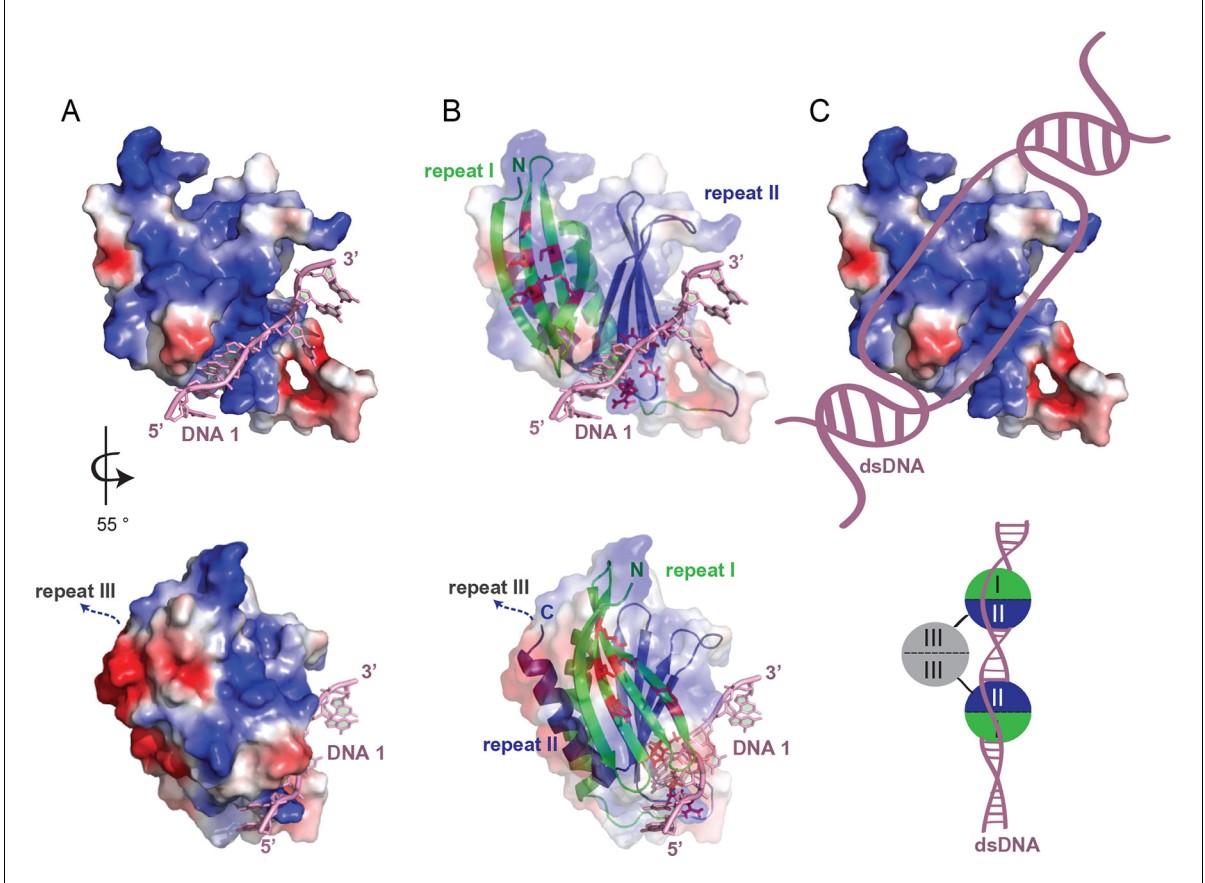

**Figure 7.** Model for unwinding of dsDNA by full-length Pur-alpha. (**A**, top) Electrostatic surface model of Pur-alpha repeat I-II in complex with one ssDNA molecule (pink). Red and blue colorations of the surface indicate negative and positive electrostatic potentials, respectively. (**B**, top) Cartoon shows in addition the structure model of PUR repeat I (green) and II (blue). DNA interaction sites, seen in the crystal structure, are shown as red sticks and correspond to the residues highlighted in *Figure 2A*. (**C**, top) Model showing the most likely overall trajectory of dsDNA (pink) when bound to Pur-alpha repeat I-II. The double-strand is locally unwound and the two separated strands bind to the two opposing binding sites on the protein. (**A**, **B**, bottom) Representation as in (**A**, **B**, top), additionally showing the C-terminus connecting to PUR repeat III. PUR repeat III likely arranges at the opposing site of the nucleic-acid-binding region. (**C**, bottom) Schematic drawing of an intermolecular Pur-alpha dimer bound to dsDNA (pink). PUR repeat III (grey) mediates dimerization, potentially orienting both nucleic-acid-binding domains (repeat I, green and II, blue) to the dsDNA. There both PUR domains could unwind larger regions of the DNA.

## Materials and methods

### Protein expression and purification

*Escherichia coli* BL21 (DE3) cells transformed with pGEX-6P-1::Pur-alpha fragments were grown at 37°C in LB medium supplemented with 100 µg/ml ampicillin. For [15]N-labeling of protein cells were grown in M9 minimal medium supplemented with 0.5 g/l [15]NH₄Cl. For selenomethionine-substituted protein, cells were grown in M9 minimal medium supplemented with an amino-acid mix of L-alanine, L-arginine, L-aspartic acid, L-cysteine, L-glutamate, L-glycine, L-histidine, L-isoleucine, L-leucine, L-lysine, L-phenylalanine, L-proline, L-serine, L-threonine, L-tyrosine, L-valine, and selenomethionine (100 mg/l each).

After reaching an $OD_{600}$ of 0.8, cell cultures were cooled down to 18°C and expression was induced by adding 0.25 mM IPTG. Cells were harvested after 18 hr of expression. GST-tagged proteins were purified by GST-affinity chromatography (GE Healthcare, Munich, Germany). After protease cleavage, the GST tag was removed by a glutathione-sepharose column. Nucleic acids were removed by using an anion-exchange Q column (GE Healthcare) followed by size exclusion chromatography with buffer containing 250 mM NaCl, 20 mM Hepes pH 8.0. For cysteine-containing and

for selenomethionine-substituted proteins 2 mM DTT was added to the buffer. For NMR experiments size exclusion chromatography was performed in 50 mM potassium phosphate buffer pH 7.0 and 200 mM NaCl (NMR buffer). Absence of nucleic-acid contamination was confirmed by measuring the ratio of absorption at 260/280 nm (*Edelmann et al., 2014*).

## Circular dichroism (CD) spectroscopy

To confirm proper protein folding of the Pur-alpha mutants CD spectra (wavelength 190–260 nm) were recorded with a JASCO-715 spectropolarimeter at 5°C in a 0.1-cm cuvette. Proteins were diluted in buffer containing 250 mM NaCl, 20 mM Hepes pH 8.0, and 2 mM DTT to a final protein concentration of 30 µM in 300 µl total volume. Five scans were taken with a speed of 50 nm/min.

## Crystallization and structure determination of *Drosophila melanogaster* Pur-alpha repeat I-II

Crystallization was carried out with freshly prepared selenomethionine-substituted Pur-alpha repeat I-II (residues 40–185) in size exclusion buffer (250 mM NaCl, 20 mM Hepes pH 8, 2 mM DTT). The protein was mixed with commercially purchased GCGGCGG ssDNA oligonucleotides, dissolved in Milli-Q $H_2O$ at a ratio 1:2.2 (protein:DNA). The final protein concentration was 1.77 mg/ml. A drop size of 3 µl and a 2:1 mixture of protein-DNA complex and crystallization solution were used for hanging-drop vapor-diffusion at 21°C using 24-well EasyXtal Crystal Support plates (Qiagen, Hilden, Germany). The crystallization solution contained 50 mM MES pH 5.2, 500 mM $(NH_4)_2SO_4$, 1 mM TCEP, and 16% PEG400. The total reservoir volume was 500 µl. Rod-shaped crystals of 160 x 20 µm size appeared within 4 days. Prior to data collection, crystals were cryoprotected in mother liquor and flash frozen in liquid nitrogen. Native dataset was recorded at 100 K at beamline ID23-2 (European Synchrotron Radiation Facility [ESRF] Grenoble, France). Crystals diffracted up to 2.0 Å. Data were integrated and scaled with XDS (*Kabsch, 1993*). Structure was solved by molecular replacement with PHASER (*McCoy et al., 2007*) using the apo-structure of *Drosophila* Pur-alpha 40–185 (PDB ID 3K44) as template and model building was manually completed using COOT (*Emsley et al., 2010*). Refinement of the native data was performed with PHENIX (*Adams et al., 2010*) using NCS and TLS. The final model was analyzed with SFCHECK (*Vaguine et al., 1999*), PHENIX, and REFMAC (*Murshudov et al., 1997;*, *Terwilliger, 2002*). Superpositioning of the apo-structure with the DNA-complexed structure of Pur-alpha was performed with the superpose algorithm (*Krissinel and Henrick, 2004*) of the program COOT. Images and movie of the crystal structure, superimpositions of the co-complex and apo-structure, as well as electrostatic surface potentials were prepared with PyMol (Version 1.7; Schrodinger LLC.; http://www.pymol.org/). All crystallographic software was used from the SBGRID software bundle (*Morin et al., 2013*). Structural model and dataset is available http://www.rcsb.org (PDB-ID: 5FGP).

## Crystallization and structure determination of *Drosophila melanogaster* Pur-alpha repeat III

Selenomethionine-substituted crystals of Pur-alpha repeat III (residues 188–258) were grown at 4°C with a protein concentration of 0.5–2 mg/ml. The crystallization solution contained 50 mM MES pH 6.5, 200 mM NaCl, 16% PEG 3350, and 6% MPD. Plate-shaped crystals of approximately 70 × 70 × 10 µm size appeared within 2–4 days. For cryo-protection, crystals were shortly incubated in reservoir solution containing 30% ethylene glycol in two steps and then flash frozen in liquid nitrogen.

Native dataset was recorded at 100 K at beamline ID14-1 [ESRF]. Crystals showed good diffraction up to 2.6 Å and belonged to space group P2₁ (see *Table 1*). The data were integrated and scaled with the XDS program package. Phases were obtained by molecular replacement using PHASER together with *Borrelia burgdorferi* Pur-alpha and *Drosophila melanogaster* Pur-alpha repeat I-II structures as a search model. Best results were achieved using a truncated version of the search models lacking the loop regions and poly-serine as amino-acid sequence. Parts of the initial model were built automatically with Buccaneer (*Cowtan, 2006*) and manually completed using COOT. Refinement was performed with PHENIX using NCS with 6 monomers per asymmetric unit. Structural model and dataset is available http://www.rcsb.org (PDB-ID: 5FGO).

## Isotopic labeling

For RNA-labeling RNase-free buffers, materials, and reagents were used. Ten picomol of chemically synthesized DNA or RNA oligonucleotides were phosphorylated at the 5'-end with 10 pmol $\gamma$-$^{32}$P ATP by T4 polynucleotide kinase (New England Biolabs, Frankfurt, Germany) with buffer A in a final volume of 20 µl. Labeling reaction was carried out at 37°C and stopped after 30 min by incubation at 70°C for 10 min. Labeled oligonucleotides were purified by a NucAway™ Spin column (Ambion, Ulm, Germany) and stored at -20°C.

## Electrophoretic mobility shift assay

The protein-nucleic acid complexes were formed in RNase-free binding buffer containing 250 mM NaCl, 20 mM Hepes pH 8.0, 3 mM MgCl$_2$, 4% glycerol, 2 mM DTT). Serial protein dilutions and a constant amount of radiolabeled nucleic acid (2.5 nM) were incubated in a total reaction volume of 20 µl for 20 min at 21°C. DNA-binding experiments contained 25 µg/ml Salmon Sperm DNA, and RNA-binding experiments contained 100 µg/ml yeast tRNA competitor. Ten microliter of the reactions were loaded onto 6% TBE polyacrylamide gels. After electrophoresis (45 min, 100 V), gels were incubated for 15 min in fixing solution ([v/v] 10% acetic acid, [v/v] 30% methanol), dried in a gel dryer (BioRad, Munich, Germany) and analyzed with radiograph films in a Protec Optimax developer (Hohmann, Hannover, Germany).

Sequences of oligonucleotides were as follows: MF0677 ssDNA/RNA, 5'-GGAGGTGGTGGAGG-GAGAGAAAAG-3'; CGG ssDNA/RNA, 5'-(CGG)$_8$–3'.

## Fluorescence-polarization experiments

For fluorescence-polarization measurements, protein-nucleic acid complexes were formed in buffer containing 500 mM NaCl, 20 mM Hepes pH 7.5, 3 mM MgCl$_2$, 2 mM DTT). In comparison to EMSA, higher salt concentrations were used (500 mM versus 250 mM) to allow for binding experiments at higher protein concentrations without aggregation of Pur-alpha. Serial protein dilutions and a constant amount of fluorescein-labeled MF0677 ssDNA or ssRNA (100 nM) were incubated for 20 min at 21°C in a total reaction volume of 40 µl. DNA-binding reactions contained 25 µg/ml Salmon Sperm DNA and RNA-binding reactions contained 100 µg/ml yeast tRNA as competitor. Measurements were performed on an Envision Multilabel reader (Perkinelmer). The excitation and emission wavelengths were 485 nm and 535 nm, respectively. The dissociation constant was calculated by fitting the data with the one-site binding model included in the program origin (OriginLab). The experiment was performed as triplicates.

Equation for one-site binding: y=Bmax*x/(k1+x). y = specific binding, x = ligand concentration, Bmax = maximum specific binding, k1 = equilibrium binding constant.

## NMR experiments

All NMR spectra were recorded in NMR buffer with 5% D$_2$O at 298 K using a Bruker Avance III spectrometer equipped with a TCI cryogenic probe head, at field strengths corresponding to 900 MHz proton Larmor frequency. To study DNA/RNA binding $^1$H,$^{15}$N HSQC NMR spectra were recorded of $^{15}$N-labeled protein (50 µM) titrated with nucleic acids with different stoichiometric ratio of protein: nucleic acid (1:0.25, 1:0.5, 1:0.75, 1:1, 1:1.25, 1:1.5, 1:2.5, and 1:5). For every spectrum, 256 increments in the $^{15}$N indirect dimension with eight scans and an interscan delay of 1 s were acquired. Spectra were recorded and processed with Topspin 3.2 (Bruker) and analyzed with CCPNMR analysis (*Vranken et al., 2005*).

## Unwinding assay

Unwinding assays were carried out according to reference (*Darbinian et al., 2001*). A dsDNA substrate was prepared by annealing a complementary 18-mer oligonucleotide to a GGN motif of the M13mp18 ssDNA plasmid. The 18-mer was labeled with $\gamma$-$^{32}$P ATP. Protein dilutions were added to a constant amount of dsDNA substrate (100 ng) in binding buffer composed of 150 mM NaCl, 20 mM Hepes pH 8.0. Samples were incubated at 37°C for 1 hr. The unwinding reaction was stopped by adding SDS to a final concentration of (v/v) 0.3%. Samples were run on 9% native polyacrylamide gels in 1x TBE buffer for 150 min at 200 V. Gels were incubated for 15 min in fixing solution ([v/v]

10% acetic acid, [v/v] 30% methanol), dried and analyzed with radiograph films. The sequence of the 18-mer oligonucleotide was as follows: 5'-TCAGAGCCGCCACCCTCA-3'.

## Filter-binding assay

Filter-binding assays were performed as described (*Wong and Lohman, 1993*). Protein was titrated to a constant amount of 1 µM MF0677 ssDNA (thereof 2.5 nM radiolabeled) in a final volume of 80 µl and incubated for 20 min at 21 °C in binding buffer 150 mM NaCl, 20 mM Hepes pH 8.0. Nitrocellulose filter (Roth, Karlsruhe, Germany) was presoaked for 10 min in 0.4 M KOH followed by intensive washing with Milli-Q $H_2O$. Nitrocellulose and nylon filters (Roth) were then equilibrated in binding buffer for 15 min. Both filters (nitrocellulose, top; nylon filter, bottom) were placed into a dot-blot apparatus (BioRad). Vacuum was applied and the wells were washed once with 80-µl binding buffer before and after samples were loaded. The nitrocellulose filters were analyzed using a phosphor imager system to measure the retained radiolabeled oligonucleotides on the nitrocellulose filter. Quantification was done using the dot blot analyzer plug-in of the ImageJ 1.47v software (National Institute of Health, USA).

## *Drosophila* genetics

Transgenic flies expressing $rCGG_{90}$ repeats were obtained as previously described (*Jin et al., 2003*). The pUAST constructs were generated by cloning cDNA of full-length *Drosophila* Pur-alpha into the pUAST transformation vectors. The constructs were confirmed by DNA sequencing and then injected in a $w^{1118}$ strain by standard methods. Fly lines were grown on standard medium with yeast paste added. All crosses were performed at 25°C.

## Electron microscopy

For scanning electron microscopy (SEM) images, whole flies were dehydrated in ethanol, dried with hexamethyldisilazane (Sigma-Aldrich, Hamburg and Seezle, Germany), and analyzed with an ISI DS-130 LaB6 SEM/STEM microscope.

## Acknowledgements

We thank Marietta Truger and Stephane Roche for support during structure determination.

## Additional information

### Funding

| Funder | Grant reference number | Author |
|---|---|---|
| Deutsche Forschungsgemeinschaft | SFB646 | Dierk Niessing |
| Deutsche Forschungsgemeinschaft | NI1110/4-1 | Janine Weber Dierk Niessing |
| Deutsche Forschungsgemeinschaft | FOR855 | Dierk Niessing |
| Deutsche Forschungsgemeinschaft | FOR2333 | Dierk Niessing |
| Bayerisches Staatsministerium für Bildung und Kultus, Wissenschaft und Kunst | Bavarian Molecular Biosystems Research Network | Tobias Madl |
| Deutsche Forschungsgemeinschaft | Emmy Noether program | Tobias Madl |
| Deutsche Forschungsgemeinschaft | MA5703/1-1 | Tobias Madl |
| Deutsche Forschungsgemeinschaft | Center for Integrated Protein Science Munich (CIPSM) | Tobias Madl |

The funders had no role in study design, data collection and interpretation, or the decision to submit the work for publication.

## Author contributions

JW, Conception and design, Acquisition of data, Analysis and interpretation of data, Drafting or revising the article, Contributed unpublished essential data or reagents; HB, CH, ZW, Acquisition of data, Analysis and interpretation of data, Contributed unpublished essential data or reagents; AW, Conception and design, Acquisition of data, Analysis and interpretation of data, Contributed unpublished essential data or reagents; RJ, TM, PJ, Analysis and interpretation of data, Drafting or revising the article, Contributed unpublished essential data or reagents; DN, Conception and design, Analysis and interpretation of data, Drafting or revising the article, Contributed unpublished essential data or reagents

## Author ORCIDs

Dierk Niessing, http://orcid.org/0000-0002-5589-369X

## Additional files

### Major datasets

The following datasets were generated:

| Author(s) | Year | Dataset title | Dataset URL | Database, license, and accessibility information |
|---|---|---|---|---|
| Weber J, Janowski R, Niessing D | 2015 | Crystal structure of D. melanogaster Pur-alpha repeat I-II in complex with DNA | http://www.rcsb.org/pdb/search/structid-Search.do?structureId=5FGP | Publicly available at the RCSB Protein Data Bank (Accession no: 5FGP). |
| Windhager A, Janowski R, Niessing D | 2015 | Crystal structure of D. melanogaster Pur-alpha repeat III | http://www.rcsb.org/pdb/search/structid-Search.do?structureId=5FGO | Publicly available at the RCSB Protein Data Bank (Accession no: 5FGO). |

**Reporting standards:**Standard used to collect data: Data were collected and uploaded according to the guidelines from the Protein Data Bank (PDB; http://www.rcsb.org/). PDB-validation reports of both structures have been uploaded to eLife as part of this submission.

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
