## [Decision Letter]

Thank you for submitting your work entitled "Structural basis of nucleic-acid recognition and double-strand unwinding by the essential neuronal protein Pur-alpha" for consideration by *eLife*. Your article has been favorably evaluated by Richard Aldrich (Senior Editor), Karsten Weis (Reviewing Editor) and by two peer reviewers.

The reviewers have discussed the reviews with one another and the Reviewing editor has drafted this decision to help you prepare a revised submission.

Summary:

In this manuscript, Weber et al. report a 2.0 Å crystal structure of the Pur-alpha repeat I-II in complex with a ssDNA oligo, and a 2.6 Å crystal structure of the Pur-alpha repeat III. In addition, the authors biochemically characterize the nucleic acid binding properties and the unwinding activity of Pur-alpha and its repeats. These in vitro data are complemented with work performed in *Drosophila*.

There was agreement amongst the reviewers that this study is of general interest and that, in general, the conclusions are well-founded. However, there were some concerns about the functional analysis of the protein and revisions are needed before the paper can be accepted for publication in *eLife*.

Essential revisions:

1) There is a concern regarding the discussion of the nucleic acid specificity of Pur-alpha: "Together, our biochemical, NMR, and x-ray crystallographic insights confirm that Pur-alpha binds DNA and RNA in the same way and thus will interact equally with both types of nucleotides (should be nucleic acids) in the cell." The authors show that the Pur-alpha repeats I-II do not display any specificity towards ssRNA or ssDNA in vitro. Furthermore, they detected only a weak binding of repeat III to nucleic acids. However, the authors did not test the full-length protein in their assays. There might be a significant contribution of the additional parts of the protein (i.e. the Gly-rich N-terminus and the Gln/Glu-rich C-terminus) towards nucleic acid specificity. Furthermore, repeat I-II together with repeat III in the context of the full-length protein might show nucleic acid specificity.

In the event that the authors cannot express the full-length protein and test it in their assays, the discussion needs be written more cautiously, and it should be made clearer what conclusions could be drawn concerning full-length Pur-alpha.

2) It remains somewhat confusing whether the nucleic acid binding activity of the mutants correlates with their abilities to rescue the neurodegeneration phenotype. The nucleic acid binding of the rescuing RI-RII mutation should be tested in vitro and compared to the affinity of non-rescuing mutations. To this end, EMSAs should be quantified and apparent KD values should be extracted.

3) DNA binding of the Pur-alpha repeats I and II occurs via largely equivalent surfaces. However, most of the residues that directly contact DNA are non-equivalent (except for the equivalent K61 in repeat I and K138 in repeat II). Mutating residues K61, N63 and R65 in repeat I (triple mutant KNR I) leads to deficiency in nucleic acid binding and unwinding. However, only K61 of this motif directly binds DNA. The contributions of these three residues to DNA binding should be discussed in more detail (direct effects by altering K61, indirect or no effects by altering N63 and R65?). To allow better comprehension/interpretation of the effects of the KNR I triple mutant, the authors should provide an additional figure panel that shows the positions and conformations of all three residues with respect to bound DNA.

4) Table 1 needs to be corrected, as it contains a few careless mistakes:

In the data collection part, the numbers for the resolution do not seem to be fully correct. The data on the Pur-alpha repeat III range from 50 to 0 Å, while the highest resolution shell is indicated as ranging from 47.7-2.6 Å, which does not correlate with the statistical values, such as I/σ etc. Furthermore, it is odd that the completeness of the data is greater for the highest resolution shell than for the full data set.

Similarly, the highest resolution shell for the Pur-alpha I-II/DNA complex is mis-indicated as ranging from 41.9 – 2.0 Å.

Although described in the Methods section, the stereochemistry outliers should be included in the table.

---

## [Author Response]

*Essential revisions: 1) There is a concern regarding the discussion of the nucleic-acid specificity of Pur-alpha: "Together, our biochemical, NMR, and x-ray crystallographic insights confirm that Pur-alpha binds DNA and RNA in the same way and thus will interact equally with both types of nucleotides (should be nucleic acids) in the cell." The authors show that the Pur-alpha repeats I-II do not display any specificity towards ssRNA or ssDNA in vitro. Furthermore, they detected only a weak binding of repeat III to nucleic acids. However, the authors did not test the full-length protein in their assays. There might be a significant contribution of the additional parts of the protein (i.e. the Gly-rich N-terminus and the Gln/Glu-rich C-terminus) towards nucleic-acid specificity. Furthermore, repeat I-II together with repeat III in the context of the full-length protein might show nucleic acid specificity. In the event that the authors cannot express the full-length protein and test it in their assays, the discussion needs be written more cautiously, and it should be made clearer what conclusions could be drawn concerning full-length Pur-alpha.*

As suggested, we expressed full-length Pur-alpha and quantified its binding to ssDNA and ssRNA using fluorescence-polarization experiments. This information is included in Figure 1 and explained in the second paragraph of the subsection “Pur-alpha binds RNA and DNA with similar affinities”.

The results show that both types of nucleic acids are bound by full-length Pur-alpha in the same affinity range. MF0677 ssRNA is bound about 2-fold stronger than ssDNA (K_D_ for DNA = 1.4 µM and for RNA = 0.7 µM), suggesting that sequences outside the PUR repeats I-II moderately contribute to the binding.

*2) It remains somewhat confusing whether the nucleic-acid-binding activity of the mutants correlates with their abilities to rescue the neurodegeneration phenotype. The nucleic-acid binding of the rescuing R I - R II mutation should be tested in vitro and compared to the affinity of non-rescuing mutations. To this end, EMSA should be quantified and apparent K_D_ values should be extracted.*

Because quantification of EMSA is usually not very accurate, in particular when more than one shifted band is observed (please see Figure 1, Figure 3, and Figure 4), we decided to perform fluorescence-polarization experiments instead. Binding experiments were performed with wild-type Pur-alpha I-II and with repeat III, as well as with all requested mutants. The results are included as a new table in Figure 3 and representative binding curves of all proteins are shown in Figure 3—figure supplement 2. The observed K_D_ values are consistent with our results from EMSA. Interestingly, mutations in PUR repeat II have stronger effects than mutations in PUR repeat I. This observation is consistent with the co-structure, in which also repeat II makes the main interactions with DNA. As shown in a previous publication (Graebsch et al. PNAS 2009), the R I – R II mutant failed to bind DNA in vitro. In summary, the quantifications provide strong support for our conclusions drawn in the initial manuscript.

We are discussing these results and their relation to in vivo rescue experiments in the Discussion.

*3) DNA binding of the Pur-alpha repeats I and II occurs via largely equivalent surfaces. However, most of the residues that directly contact DNA are non-equivalent (except for the equivalent K61 in repeat I and K138 in repeat II). Mutating residues K61, N63 and R65 in repeat I (triple mutant KNR I) leads to deficiency in nucleic-acid binding and unwinding. However, only K61 of this motif directly binds DNA. The contributions of these three residues to DNA binding should be discussed in more detail (direct effects by altering K61, indirect or no effects by altering N63 and R65?). To allow better comprehension/interpretation of the effects of the KNR I triple mutant, the authors should provide an additional figure panel that shows the positions and conformations of all three residues with respect to bound DNA.*

We address this issue with a more detailed description and discussion of these residues in the Results and Discussion sections. As suggested, we also provide an additional figure, in which residues K61, N63, and R65 are shown as close-up (Figure 2—figure supplement 4). They can now be directly compared with the close-up of residues K138, N140, and R142 in Figure 2. We are grateful for this suggestion, as this aspect was indeed not sufficiently covered.

*4) Table 1 needs to be corrected, as it contains a few careless mistakes: In the data collection part, the numbers for the resolution do not seem to be fully correct. The data on the Pur-alpha repeat III range from 50 to 0 Å, while the highest resolution shell is indicated as ranging from 47.7-2.6 Å, which does not correlate with the statistical values, such as I/σ etc. Furthermore, it is odd that the completeness of the data is greater for the highest resolution shell than for the full data set.*

*Similarly, the highest resolution shell for the Pur-alpha I-II/DNA complex is mis-indicated as ranging from 41.9* – *2.0 Å.*

*Although described in the Methods section, the stereochemistry outliers should be included in the table.*

We are grateful to the reviewers for pointing us to these obvious errors. We have corrected and double-checked all numbers, and apologize for these mistakes.

Regarding the “greater completeness of the data for the highest resolution shell than for the full data set”, we would like to emphasize that this is in fact correct. The likely reason is that low-resolution data are less complete than the average dataset. Since low-resolution data have a considerable impact on crystallographic statistics, effects as in our datasets can sometimes be observed. We would like to add that we verified the correctness of these statistics (including the identical redundancies for the dataset of Pur-alpha repeat III).

As suggested, we moved the statistics of the stereochemistry from the Methods section to Table 1 and provide the wavelength of data collection in this table. We also added information on the beamline, detector distance, number of images, and oscillation range. We hope that this table now includes all relevant data.